# Facile Approach for the Potential Large-Scale Production of Polylactide Nanofiber Membranes with Enhanced Hydrophilic Properties

**DOI:** 10.3390/ma16051784

**Published:** 2023-02-21

**Authors:** Changmei Jiang, Yuan Tian, Luolan Wang, Shiyou Zhao, Ming Hua, Lirong Yao, Sijun Xu, Jianlong Ge, Gangwei Pan

**Affiliations:** 1National & Local Joint Engineering Research Center of Technical Fiber Composites for Safety and Protection, School of Textile and Clothing, Nantong University, Nantong 226019, China; 2Chinatesta Textile Testing Services (Zhejiang), Shaoxing 312000, China

**Keywords:** polylactide, cellulose diacetate, nanofiber membrane, oil–water separation, hydrophilic property

## Abstract

Polylactide (PLA) nanofiber membranes with enhanced hydrophilic properties were prepared through electrospinning. As a result of their poor hydrophilic properties, common PLA nanofibers have poor hygroscopicity and separation efficiency when used as oil–water separation materials. In this research, cellulose diacetate (CDA) was used to improve the hydrophilic properties of PLA. The PLA/CDA blends were successfully electrospun to obtain nanofiber membranes with excellent hydrophilic properties and biodegradability. The effects of the additional amount of CDA on the surface morphology, crystalline structure, and hydrophilic properties of the PLA nanofiber membranes were investigated. The water flux of the PLA nanofiber membranes modified with different CDA amounts was also analyzed. The addition of CDA improved the hygroscopicity of the blended PLA membranes; the water contact angle of the PLA/CDA (6/4) fiber membrane was 97.8°, whereas that of the pure PLA fiber membrane was 134.9°. The addition of CDA enhanced hydrophilicity because it tended to decrease the diameter of PLA fibers and thus increased the specific surface area of the membranes. Blending PLA with CDA had no significant effect on the crystalline structure of the PLA fiber membranes. However, the tensile properties of the PLA/CDA nanofiber membranes worsened due to the poor compatibility between PLA and CDA. Interestingly, CDA endowed the nanofiber membranes with improved water flux. The water flux of the PLA/CDA (8/2) nanofiber membrane was 28,540.81 L/m^2^·h, which was considerably higher than that of the pure PLA fiber membrane (387.47 L/m^2^·h). The PLA/CDA nanofiber membranes can be feasibly applied as an environmentally friendly oil–water separation material because of their improved hydrophilic properties and excellent biodegradability.

## 1. Introduction

Given that the problem of environmental pollution caused by oil leaks and irregular industrial sewage emissions has had a considerable influence on the human living environment in recent years, the treatment of oil-containing wastewater has garnered considerable interest [1,2]. Researchers have committed themselves to the creation of novel materials to combat the environmental pollution caused by oil-containing wastewater [3,4]. In addition, several physical and chemical methods, such as the gravity method, floating method, centrifugal method, adsorption method, and electrochemical treatment, are used to separate oil and water mixes [5]. However, these traditional methods have the problems of low separation efficiency, high energy consumption, and secondary pollution emission [6,7]. Therefore, creating a highly selective and efficient advanced separation method to address these issues is of the utmost importance [8].

As a reasonably advanced purification and separation method [9], membrane technology has the benefits of continuous operation, high efficiency, and low energy consumption and is therefore suitable for treating various types of industrial effluents. Electrospun technology is widely used in the preparation of nanofiber membranes because electrospun nanofiber membranes are formed via fiber stacking and thus have high surface area-to-pore ratios [10,11,12], allowing for great fiber membrane penetration during oil and water separation [13,14,15].

The majority of existing membrane materials are nonbiodegradable and can therefore readily contribute to secondary pollution. Utilizing biodegradable polymers can circumvent this problem. The biochemical, nontoxic, biocompatible, and regenerative properties of polylactide (PLA) have attracted extensive interest [16]. PLA is a biodegradable polyester that is obtained from renewable resources [17] such as corn, starch, and sugar cane [18]. However, PLA has poor hydrophilicity [19]. It must be hydrophilic when used as an oil–water separation material to achieve superior oil resistance, antipollution property, and high separation efficiency [20]. Cellulose diacetate (CDA) is an environmentally friendly and biodegradable modified cellulose material [21]. It is one of the most applicable polymers for nanofiber membranes due to its high hydrophilicity, good solubility, and low cost in multiple solvent systems [22].

Current methods for the hydrophilic modification of PLA are classified into subject modification and surface modification [23]. Qin et al. [24] prepared a PLA/carbon nanotubes (CNT_S_) fiber membrane through electrospinning then added a PLA/SiO_2_ hydrophilic membrane on one side of the PLA/CNT_S_ fiber membrane via electrospinning to obtain a Janus–PLA fiber membrane with excellent separation efficiency and mechanical performance. The membrane can also separate oil–water emulsion solutions. The Janus–PLA fiber membrane has promising applications in the treatment of oil-containing wastewater. The extremely thin PLA electrospun membrane was hot-pressed on the PLA nonwoven microfiber membrane. Then, the hydrophilic compound polydopamine was coated onto the surface. By modifying the surface of the PLA membrane, the structural stability and surface hydrophilicity of the membrane were enhanced. The modified PLA fiber membrane presented excellent comprehensive separation properties for an oil-in-water emulsion with surfactant under a wide range of pressures [25]. Although the performance of the hydrophilic PLA membrane in previous research met the requirements of oil–water separation, the process of membrane preparation is too complicated and unsuitable for industrial production. Moreover, the poor fastness of the hydrophilic agent, such as polydopamine, on the surface of the PLA membrane is a crucial problem that needs resolution.

The purpose of this study is to develop a simple and stable PLA oil–water separation membrane that can be fully biodegradable and does not involve any nonbiodegradable additives and surface modifications. The effect of the amount of CDA on the structure and properties of the PLA nanofiber membranes was studied. Therefore, PLA/CDA nanofiber membranes were prepared with different PLA/CDA blending ratios. First, the morphology and tensile properties of the PLA nanofiber membranes were measured by using a scanning electron microscope (SEM) and fiber strength tester. Second, the chemical and crystalline structures of the PLA membranes were observed by using Fourier transform infrared spectroscopy (FTIR) and X-ray diffraction (XRD). Finally, the water contact angle and water flux of the PLA membranes were tested to demonstrate the application potential of the PLA nanofiber membranes in oil–water separation.

## 2. Materials and Methods

The PLA (6202D, with an average molecular weight of 1.6 × 10^5^) utilized in this study was manufactured by NatureWorks Co., Ltd. (Minnetonka, MN, USA). CDA (acetyl 2.45) was provided by Nantong Cellulose Fibers Co., Ltd. (Nantong, China). Analytical-grade chloroform (CHL) was purchased from Shanghai Lingfeng Chemical Reagent Co., Ltd. (Shanghai, China). Analytical-grade *N*,*N*-dimethylformamide (DMF) was purchased from Jiani Chemical Co., Ltd. (Wuxi, China).

First, PLA or CDA was dissolved in CHL/DMF (*w*/*w*: 9/1) solvent at room temperature and stirred with a magnetic mixer for 24 h to form pure PLA spinning solutions with 5–8 wt% concentration. The 8 wt% (preferably based on the previous experiment) PLA/CDA spinning solutions with different PLA/CDA blending ratios were prepared in the same way. The blending ratios of PLA/CDA were 9/1, 8/2, 7/3, and 6/4.

The PLA nanofiber membrane was fabricated through electrospinning at the temperature of 25 °C ± 3 °C and relative humidity of 50% ± 5%. The electrospinning settings were set as follows: spinning voltage of 20 kV, instilling speed of 1 mL/h, distance from the tip to the collector of 20 cm, and receiving roller winding speed of 65–80 r/min. The above-mentioned solution was used under the same process conditions for the preparation of electrospinning to obtain pure PLA, pure CDA, PLA/CDA (9/1), PLA/CDA (8/2), PLA/CDA (7/3), and PLA/CDA (6/4) fiber membranes.

All samples were observed by using SEM (Gemini SEM 300, Oberkochen, Germany). Each specimen was coated with gold for 60 s before SEM observation.

A Nicolet IS10 spectrometer (Thermo Fisher, Waltham, MA, USA) was used to measure the fiber membranes with an average of 32 scans over the range of 600–4000 cm^−1^.

XRD patterns were recorded by using an X-ray diffractometer (Rigaka, Ultima IV, Tokyo, Japan) with CuKα radiation (λ = 0.1542 nm) in the range of 5–50°. The scanning voltage was 40 kV, the scanning current was 40 mA, and the scanning speed was 3°/min.

The tensile properties of the membranes were tested by using fiber strength tester instruments (XQ-2, Shanghai New Fiber Instrument Co., Ltd., Shanghai, China). Experimental parameters and sample standards were as follows: the stretching speed was 10 mm/min, the clamping distance was 30 mm, the pre-tension was 0.1 cN, and the width and length of the sample were 3 mm and 50 mm, respectively. After calculating the tensile strength of the membranes, the stress–strain curves of the membranes were obtained.

Contact angle was measured by using contact angle measuring instruments (WCA/OCA, Ding Sheng, SDC-350, Suzhou, China). Deionized water (3 μL) was deposited onto the surface of each test sample. All measurements were performed three times, and the average value was calculated.

The PLA fiber membrane was fixed between a conical bottle and a glass tube and then moistened with 1 mL of deionized water for 30 s. The deionized water was poured onto the surface of the membrane from the glass tube. During this process, the liquid level was kept at 10 cm, which corresponded to a separation pressure of 1 kPa. Water flux filtering device was shown in Figure 1. Infiltration flux was calculated in accordance with Formula (1):*J* = *V*/(*A* × *T*).(1)

In the formula, *V* is the penetration volume, *A* is the effective separation area, and *T* is the penetration time.

## 3. Results and Discussion

### 3.1. Morphology of the PLA Fiber Membranes

PLA electrospinning experiments were conducted with PLA spinning solution concentrations of 5 wt%, 6 wt%, 7 wt%, and 8 wt% to determine the suitable spinning concentrations of the PLA fiber membranes. Figure 2 shows that the fibers have changed from numerous beaded structures (Figure 2a) into uniform fibers (Figure 2b); as the concentration of PLA spinning solution is increased, the distribution of the diameter of the fibers becomes increasingly uniform. Figure 2 illustrates that fiber diameter increases with the increase in spinning solution concentration, likely because with the increase in PLA content, the viscosity of the solution increases gradually (Figure 3). Finally, the distribution of the diameter of the fibers determined through SEM shows that the diameter of the 8 wt% PLA fiber membrane has more a uniform distribution than the other membranes.

We conducted multiple experiments with PLA/CDA blending ratios of 9/1, 8/2, 7/3, and 6/4 to investigate the influence of PLA/CDA blending ratios on the structure and hydrophilicity of the PLA fiber membranes. Figure 4 shows that as the CDA proportion in the fiber membrane is increased, the diameter of the fiber decreases, beading intensifies, and the uniformity of the fiber membrane deteriorates. As shown in the upper right corner of the high-power SEM micrographs, the surface of the fiber membrane containing PLA has uniformly dense pores resulting from the use of chloroform (CHL) as a solvent, which evaporates rapidly during electrospinning.

### 3.2. Infrared Spectrum Analysis

Figure 5 shows the FTIR spectra of the fiber membranes of pure CDA, PLA, and PLA/CDA with different proportions. Pure CDA has a broad absorption band at 3500 cm^−1^ that can be attributed to the stretching vibration of −OH [26]. Compared to other PLA/CDA and PLA membranes, PLA/CDA (6/4) membranes show an obvious absorption band at 3500 cm^−1^. The increase in the proportion of CDA is conducive to improving the hydrophilicity of PLA. The characteristic peak of the PLA and PLA/CDA nanofiber membranes is located at 2945 cm^−1^ and represents the asymmetry and stretching of C–H. The CDA and PLA membranes have an obvious peak at approximately 1736 cm^−1^, indicating that they include C=O. The band at 1736 cm^−1^ of pure PLA is stronger than that of pure CDA. PLA has a −CH_3_ stretching vibration at 1453 cm^−1^. The pure CDA fiber membrane also has a −CH_3_ stretching vibration at 1433 cm^−1^. With the increase in CDA content, the band at 1433 cm^−1^ of the PLA/CDA nanofiber membrane gradually weakens. The pure PLA fiber membranes show the asymmetric deformation vibration of C–O–C at 1179 cm^−1^. The characteristic peak at 1078 cm^−1^ is assigned to C–O stretching and vibration [27] and weakens gradually as the CDA concentration is increased.

### 3.3. Crystal Structure Analysis

The XRD curves of the PLA, CDA, and PLA/CDA fiber membranes (Figure 6) show typical crystal diffraction peaks for the pure PLA fiber membranes and PLA/CDA (9/1) nanofiber membranes, but no obvious diffraction peaks for other PLA/CDA fiber membranes. The existence of CDA affects the crystallization of PLA, resulting in poor crystallization behavior. The stacking of the CDA macromolecular chain with increasing CDA content disturbs the arrangement of PLA macromolecular chains, worsening the crystallization behaviors of PLA/CDA (8/2) and PLA/CDA (7/3). Moreover, with the further increase in CDA concentration, the arrangement of the PLA/CDA (6/4) molecular chain becomes significantly influenced by CDA. Therefore, the crystal diffraction peaks of PLA/CDA (6/4) are almost similar to those of CDA. Figure 6 illustrates that the crystalline structure of the PLA/CDA fiber membranes has not greatly improved.

### 3.4. Mechanical Property Analysis

The effect of CDA concentrations on the tensile properties of PLA membranes was investigated. As a result of the poor compatibility between PLA and CDA, as illustrated in Figure 7, the beaded structure of the PLA nanofiber membrane increases progressively with the increase in CDA content. Owing to the random distribution of beaded structures within the nanofiber membrane, tensile strength decreases as the CDA content is increased. The tensile strength of the PLA/CDA (6/4) fiber membrane is slightly higher than that of the PLA/CDA (8/2) and PLA/CDA (7/3) nanofiber membranes. The PLA/CDA (6/4) nanofiber membrane has slightly higher strength than the PLA/CDA (8/2) and PLA/CDA (7/3) nanofiber membranes likely because it contains numerous beaded structures that are difficult to break because they are distributed in sheets. Additionally, compared with the PLA/CDA (8/2) and PLA/CDA (7/3) nanofiber membranes, the PLA/CDA (6/4) fiber membrane has better crystallization and therefore slightly higher tensile strength.

### 3.5. Water Contact Angle Analysis

The effect of amounts of CDA addition on the hydrophilicity of the membrane surface was investigated by comparing the water contact angle of the pure PLA fiber membrane with that of the modified PLA fiber membrane. Figure 8a illustrates that the pure PLA membrane is hydrophobic and has a contact angle of approximately 134.90°. Theoretically, the wetting of liquid droplets on a solid surface is mainly determined by chemical components and geometric microstructures [28]. Figure 8f demonstrates that the pure CDA fiber membrane has a contact angle of 85.71°, indicating that it is hydrophilic. In this study, the hydrophilicity of the PLA fiber membranes is improved by the addition of the hydrophobic compound CDA. Figure 8b–e shows that as the CDA amount is increased, the water contact angle of the PLA mixed fiber membrane gradually decreases in concurrently with the gradual increase in hydrophilicity. The contact angle of the PLA/CDA (6/4) fiber membrane has decreased to 97.82°. The contact angle results show that the combination of CDA and PLA may effectively increase the hydrophilicity of the fiber membrane.

### 3.6. Water Flux Analysis

Figure 9 illustrates the water flux results of different PLA fiber membranes under the pressure of 1 MPa. It shows that the water flux of the pure PLA fiber membrane is 387.47 L/m^2^·h and that of the pure CDA fiber membrane is 151,021.34 L/m^2^·h. This is due to the hydrophobicity of PLA, which leads to the low water flux, and the CDA has better hydrophilicity, so the water flux of the CDA membrane is relatively high. The water flux of the PLA/CDA fiber membrane has improved due to the addition of CDA. The water flux of PLA/CDA (9/1) is 7259.64 L/m^2^·h and that of PLA/CDA (6/4) is 7276.06 L/m^2^·h. Hence, the water flux of PLA/CDA (9/1) is similar to that of PLA/CDA (6/4). The PLA/CDA (7/3) nanofiber membrane has a higher CDA content than the PLA/CDA (9/1) nanofiber membrane and excellent hydrophilicity. Therefore, the water flux of the PLA/CDA (7/3) nanofiber membrane is higher than that of the PLA/CDA (9/1) nanofiber membrane. The water flux of the PLA/CDA (7/3) nanofiber membrane is 9593.19 L/m^2^·h. The water flux of the PLA/CDA (6/4) fiber membrane has decreased compared with that of PLA/CDA (7/3) nanofiber membrane because PLA and CDA are not extremely compatible and the number of beaded structures on the surface of the PLA/CDA (6/4) fiber membrane has increased with the increase in the CDA content in the PLA membrane. The membrane with the highest water flux is the PLA/CDA (8/2) nanofiber membrane, because it can be seen from the SEM image that the diameter of the PLA/CDA (8/2) nanofiber membrane is thinner and uniform, which is more conducive to the passing of water. PLA/CDA (8/2) has a water flux of 28540.81 L/m^2^·h.

## 4. Conclusions

Hydrophilic PLA nanofiber membranes were successfully prepared through a relatively simple electrospinning method. The introduction of CDA improved the hydrophilicity of the PLA nanofiber membrane, and SEM revealed that the diameter of the fibers in the common PLA membrane had decreased, thereby increasing the surface area of the nanofiber membrane and further enhancing hydrophilicity. However, the crystalline structure of the PLA fiber membrane remained unchanged. Given the poor compatibility between PLA and CDA, the tensile properties of the PLA/CDA nanofiber membranes worsened. Nonetheless, the hydrophilic properties of the PLA nanofiber membranes produced through this approach significantly improved, and the water contact angle of the PLA/CDA (6/4) fiber membrane reached 97.8°. Moreover, the PLA/CDA (8/2) nanofiber membrane showed good water flux of 28,540.81 L/m^2^·h under the pressure of 1 kPa. Therefore, the modified PLA nanofiber membranes provide certain ideas for oil–water separation films. Nevertheless, CDA has limited effectiveness for enhancing the hydrophilicity of PLA. In the future, separation deposition technology for CDA will be applied to improve the hydrophilicity of PLA to obtain superhydrophilic and completely biodegradable nanofiber membrane materials for oil–water separation.

## Figures and Tables

**Figure 1 materials-16-01784-f001:**
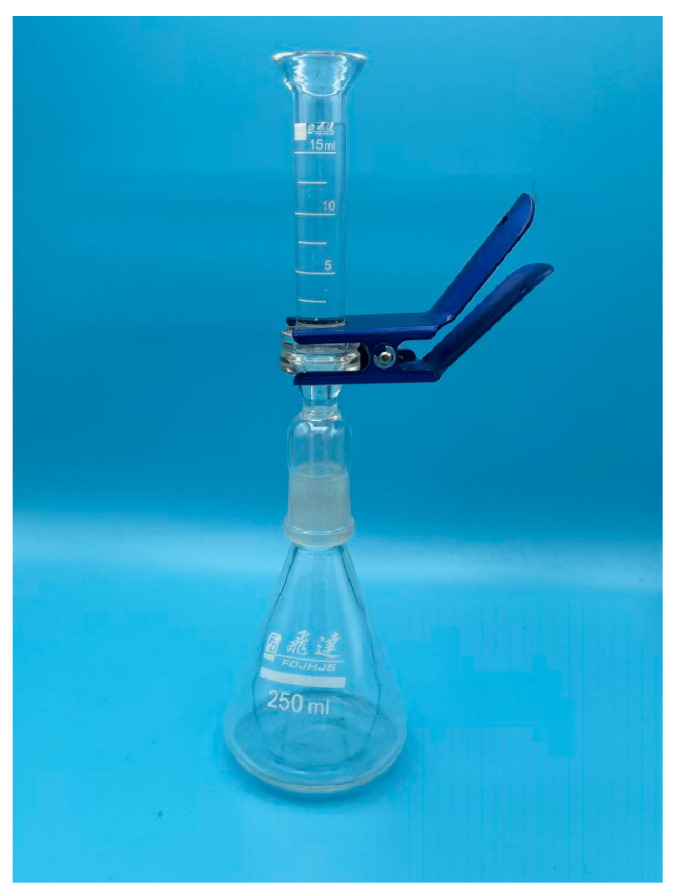
Water flux filter device diagram.

**Figure 2 materials-16-01784-f002:**
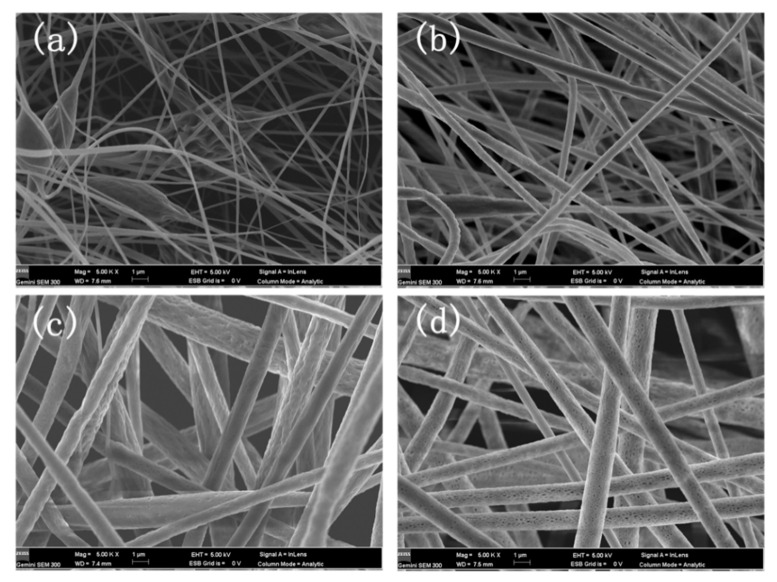
SEM micrographs of PLA fiber membranes with different concentrations: (**a**) 5 wt%, (**b**) 6 wt%, (**c**) 7 wt%, (**d**) 8 wt%.

**Figure 3 materials-16-01784-f003:**
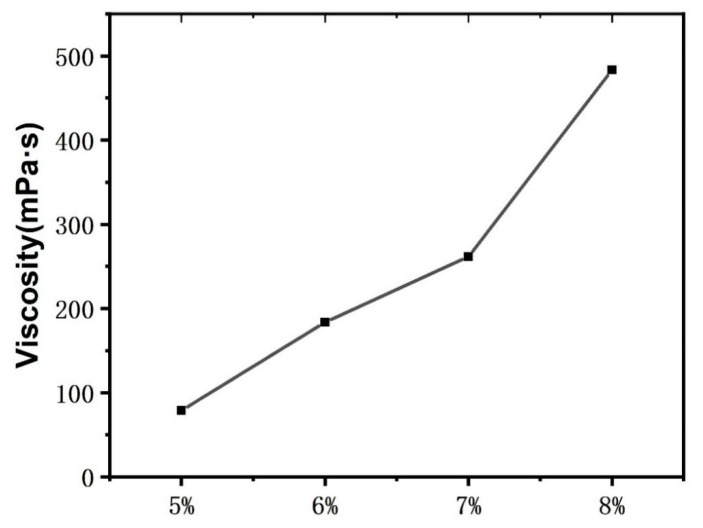
Diagram of the viscosity of the PLA spinning solutions with different concentrations.

**Figure 4 materials-16-01784-f004:**
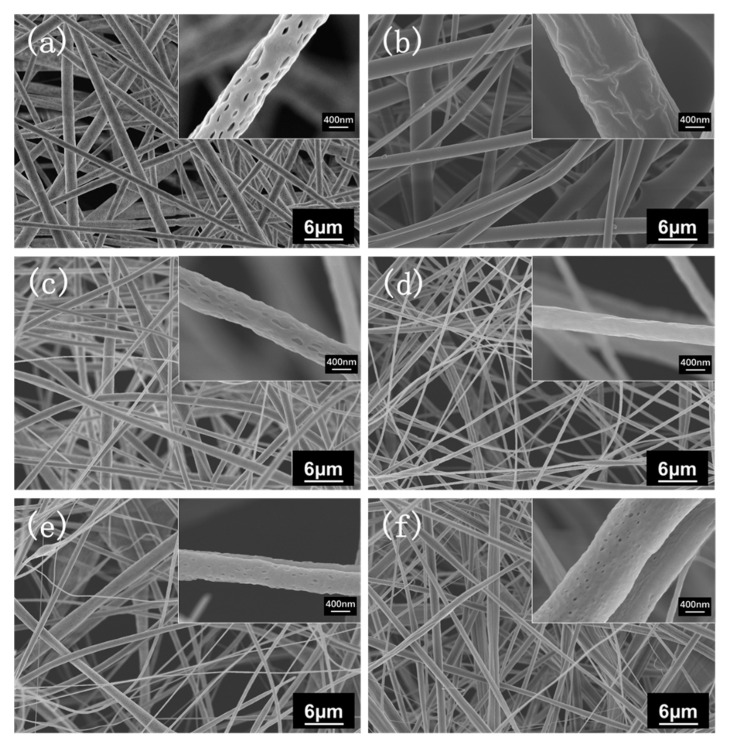
SEM micrographs of different fiber membranes: (**a**) PLA, (**b**) CDA, (**c**) PLA/CDA (9/1), (**d**) PLA/CDA (8/2), (**e**) PLA/CDA (7/3), (**f**) PLA/CDA (6/4).

**Figure 5 materials-16-01784-f005:**
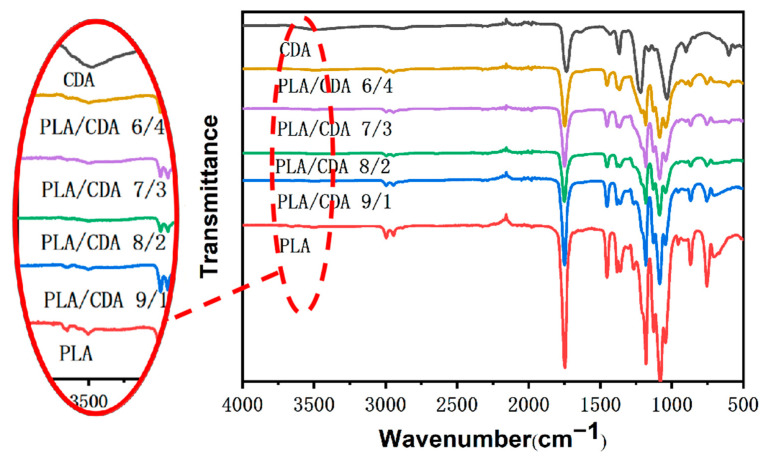
FTIR diagram of pure PLA, pure CDA, and PLA/CDA mixture fiber membranes.

**Figure 6 materials-16-01784-f006:**
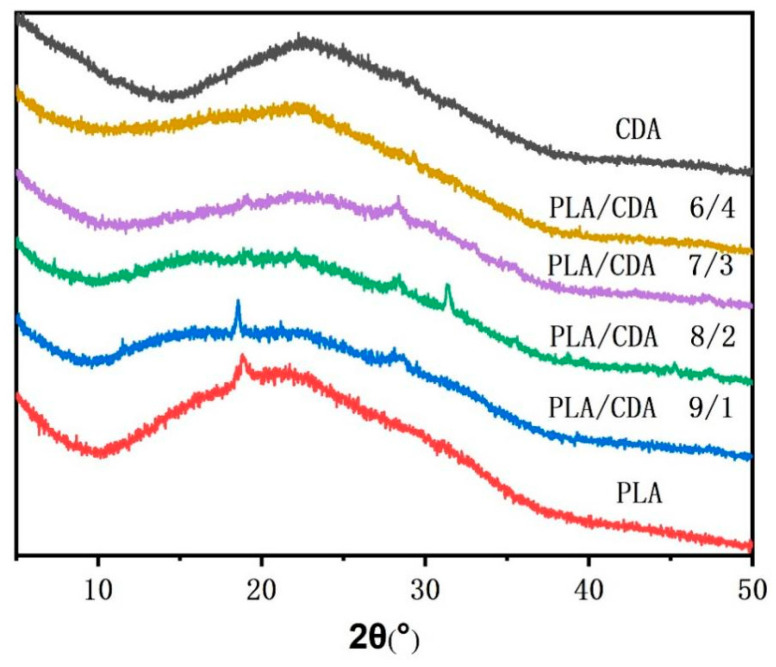
XRD of pure PLA, pure CDA, and PLA/CDA fiber membranes.

**Figure 7 materials-16-01784-f007:**
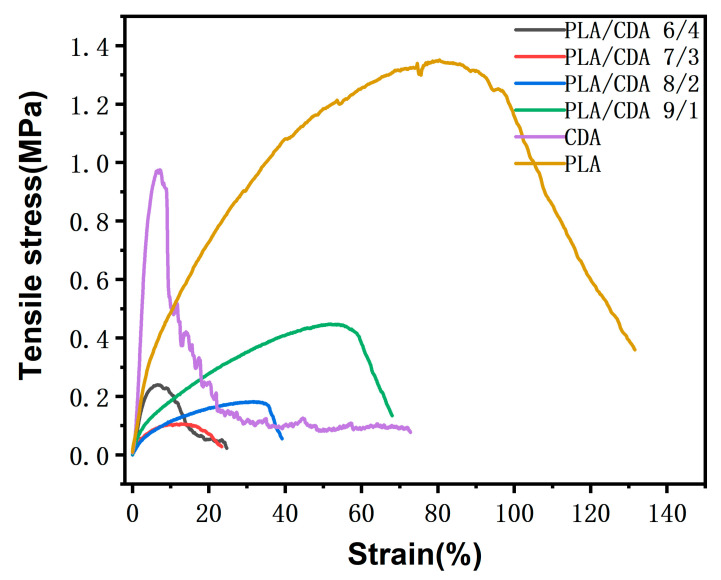
Stress–strain curves of pure PLA, pure CDA, and PLA/CDA mixed fiber membranes.

**Figure 8 materials-16-01784-f008:**
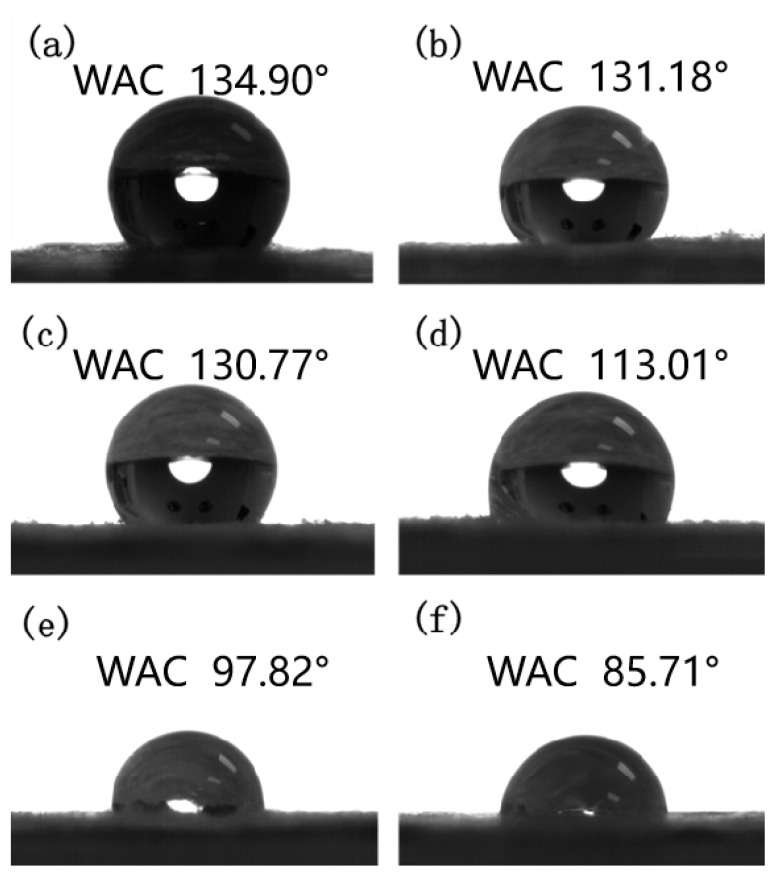
Water contact angle of different fiber membranes: (**a**) PLA, (**b**) PLA/CDA (9/1), (**c**) PLA/CDA (8/2), (**d**) PLA/CDA (7/3), (**e**) PLA/CDA (6/4), (**f**) CDA.

**Figure 9 materials-16-01784-f009:**
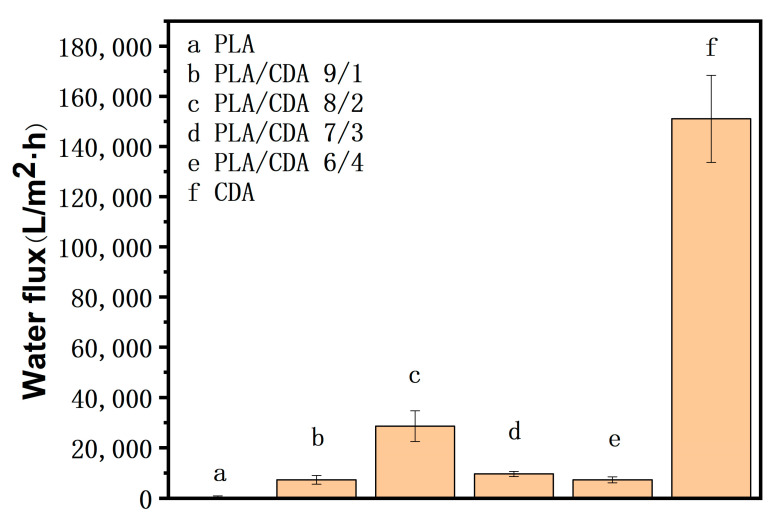
Water flux maps of different polymers: (**a**) PLA, (**b**) PLA/CDA (9/1), (**c**) PLA/CDA (8/2), (**d**) PLA/CDA (7/3), (**e**) PLA/CDA (6/4), (**f**) CDA.

## Data Availability

The raw data supporting the conclusions of this article will be made available by the authors, without undue reservation.

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
