# Peer review of "Facile Approach for the Potential Large-Scale Production of Polylactide Nanofiber Membranes with Enhanced Hydrophilic Properties"

_materials, 2023, doi:10.3390/ma16051784_

Round 1

Reviewer 1 Report

Although the article deals with interesting issues related to membranes, it requires some changes before it can be accepted for publication.

First of all, the authors should elaborate more on the description and cause of the observed research results. At present, in most of the description of the obtained results, the article resembles a research report.

Moreover:

1. The authors should extend the description of the research methodology used. In its current form, it is too laconic.

2. Why did we use such ratios of the PLA/CDA mixture in the work? Was it dictated? Earlier research or literature analysis?

3. In the methodology, the authors write that the spinning solution had a concentration of 8% by mass. However, in the results and discussion, the authors write that the spinning solution had a concentration of 5 to 8% by mass. Please unify the description.

4. On what basis did the authors conclude that xxxxx. Please better justify this statement.

5. The occurrence of the wide peak at 3500 cm-1 indicated by the authors is not reflected in Fig. 4.

6. The observed water contact angle for pure PLA is quite high for this polymer. What can the obtained value result from?

7. Since the authors claim that the developed membrane will separate the water-oil mixture well, why did they not perform this type of research.

Reviewer 2 Report

The manuscript Materials 202190875 presents the investigations regarding a method of obtaining membranes with increased hydrophilicity based on nanofibers obtained by electrospinning solutions of polylactic acid and cellulose diacetate. According to the arguments presented by the authors, the main application of these designed membranes is the separation of oil from water.

Increasing the hydrophilicity of polylactic acid nanofiber membranes was achieved by increasing the content of cellulose diacetate. Simultaneously with the increase in hydrophilicity, the effects of the addition of cellulose diacetate in the spinning solution on morphology, crystallinity, tensile strength, but especially on water permeability are investigated.

In the end it is concluded that the new PLA/CDA nanofiber membranes can be feasibly applied as an environmentally friendly oil–water separation material because of their improved hydrophilic properties and excellent biodegradability.

The introduction is well-written, because it summarizes recent research related to the topic and highlights gaps in current knowledge. The authors establish the originality of the research aims by demonstrating the need for investigations in the topic area.

The subject is not completely original, but compared to other published materials, it adds to the field of the subject an overall approach to the problems of nanofiber membranes used in the separation of oil from water

However, some aspects should be fixed to increase the quality of the manuscript as follows:

·         Abbreviations must be explained as soon as they are used, even if they are known or intuitive to readers in the field (for example: CNTs or CHL)

·         The presentation of materials and methods must be completed with information regarding viscosity determinations (apparatus, method or technique), tensile strength determinations (specimen dimensions, test speed, method or standard) and water permeability (water flux) determinations of the membranes. In this last case, even a image/drawing of the test device would be useful (as it is not a standardized method).

·         The first observation regarding the investigations of the morphology of the PLA fibers resulting from electrospinning is that at high concentrations of the spinning solution, the diameter of the fibers already exceeds the "nano" range and it is inappropriate to call them nanofibers. The authors must find a way of expression that does not affect the scientific reality demonstrated by the SEM.

·         Regarding the viscosity of solutions of different concentrations of PLA, the aspect is well known and increasing the concentration of the spinning solution to 8% actually leads to a dimensional uniformity but removes the fibers from the „nano” range.

·         In my opinion, the spinning of the cellulose diacetate solution did not produce nanofibers. (Figure 3 d)

·         A determination of the size of the fiber diameters and an analysis of their dispersion is felt necessary, otherwise the authors' assessments remain only qualitative.

·         In Figure 6 there is a mismatch between the title of the figure and the titles of the graph axes! Tensile strength is not the same as stress. The figure shows stress-strain curves but the title is wrong

·         I recommend either replacing Figure 7 (although very suggestive) with a table with the values of the contact angle or completing it by marking the value of the contact angle on the recorded images.

·         In the text it is written "the water flux of the pure PLA film is 387.47 L/m2·h and that of the pure CDA nanofiber membrane is 151 021.34 L/m2·h." It is probably a mistake not only about a film but also about a nanofiber membrane. The aspect must be corrected or explained!

·         The authors do not sufficiently explain the values obtained for the water flux (otherwise one of the most important characteristics for a membrane) for membranes made of fibers with different compositions. Why the highest water flux is for the PLA/CDA (8/2) nanofiber membrane?

The paper is well written, the text is clear and easy to read, and the conclusions are consistent with the evidence and arguments presented, but there are some cases in which the results obtained are not sufficiently explained or interpreted and this must be corrected!

Round 2

Reviewer 1 Report

The authors sufficiently answered the comments/questions contained in the first review. The article can be accepted for publication in its current form.

Author Response

We appreciate your valuable comments.

Reviewer 2 Report

In the manuscript corrected by the authors, there is still a mistake in figure 7 (former figure 6). The figure does indeed contain stress-strain curves, but the titles of the x and y axes are still wrong. The authors changed the title of the figure instead of changing the titles of the axes. I ask the authors to replace "Tensile strength" with "Tensile stress" for the y-axis (Tensile strength is a single value - the maximum value of tensile stress supported by a material that did not break), and "Percentage of elongation" with "Strain" for the x axis, and to keep the previous title of the figure "Stress–strain curves of pure PLA, pure CDA, and PLA/CDA mixed fiber membranes"

Author Response

Thank you for your comments and suggestions. We have revised the titles of axis in Figure 7.

"Figure 7. Stress–strain curves of pure PLA, pure CDA, and PLA/CDA mixed fiber membranes."